# Offensive Behavior, Striatal Glutamate Metabolites, and Limbic–Hypothalamic–Pituitary–Adrenal Responses to Stress in Chronic Anxiety

**DOI:** 10.3390/ijms21207440

**Published:** 2020-10-09

**Authors:** Enrico Ullmann, George Chrousos, Seth W. Perry, Ma-Li Wong, Julio Licinio, Stefan R. Bornstein, Olga Tseilikman, Maria Komelkova, Maxim S. Lapshin, Maryia Vasilyeva, Evgenii Zavjalov, Oleg Shevelev, Nikita Khotskin, Galina Koncevaya, Anna S. Khotskina, Mikhail Moshkin, Olga Cherkasova, Alexey Sarapultsev, Roman Ibragimov, Igor Kritsky, Jörg M. Fegert, Vadim Tseilikman, Rachel Yehuda

**Affiliations:** 1Department of Pediatric Psychiatry, Psychotherapy and Psychosomatics, University of Leipzig, 04107 Leipzig, Germany; 2Department of Medicine, Technical University of Dresden, 01309 Dresden, Germany; stefan.bornstein@uniklinikum-dresden.de; 3School of Medical Biology, South Ural State University, 454080 Chelyabinsk, Russia; chrousos@gmail.com (G.C.); diol2008@yandex.ru (O.T.); mkomelkova@mail.ru (M.K.); lapshin1982@yandex.ru (M.S.L.); carin-shik@mail.ru (M.V.); vadimed@yandex.ru (V.T.); 4University Research Institute of Maternal and Child Health and Precision Medicine, National and Kapodistrian University of Athens, 11527 Athens, Greece; 5College of Medicine, SUNY Upstate Medical University, Syracuse, NY 13210, USA; PerrySe@upstate.edu (S.W.P.); wongma@upstate.edu (M.-L.W.); licinioj@upstate.edu (J.L.); 6Rayne Institute, Division of Diabetes & Nutritional Sciences, Endocrinology and Diabetes, Faculty of Life Sciences & Medicine, Kings College London, London SE5 9PJ, UK; 7Department of Fundamental Medicine, Chelyabinsk State University, 454001 Chelyabinsk, Russia; 8Institute of Cytology and Genetics, Siberian Branch of the Russian Academy of Science, 630090 Novosibirsk, Russia; zavjalov@bionet.nsc.ru (E.Z.); shevelev.oleg.nsk@gmail.com (O.S.); khotskin@bionet.nsc.ru (N.K.); g-kon@ngs.ru (G.K.); dotcenko@bionet.nsc.ru (A.S.K.); mmp@bionet.nsc.ru (M.M.); 9Biophysics Laboratory, Institute of Laser Physics, Siberian Branch of the Russian Academy of Science, 630090 Novosibirsk, Russia; o.p.cherkasova@gmail.com; 10Institute of Natural Sciences and Mathematics, Ural Federal University named after the first President of Russia, 620026 Ekaterinburg, Russia; a.sarapultsev@gmail.com (A.S.); Ibragimovroman98@yandex.ru (R.I.); igor81218@gmail.com (I.K.); 11Institute of Immunology and Physiology, Ural Branch of the Russian Academy of Science, 620049 Ekaterinburg, Russia; 12Department of Child and Adolescent Psychiatry, Psychotherapy, and Psychosomatics, University of Ulm, 89075 Ulm, Germany; Joerg.Fegert@uniklinik-ulm.de; 13Traumatic Stress Studies Division, ICAHN School of Medicine at Mount Sinai, New York, NY 10029-6574, USA; rachel.yehuda@va.gov

**Keywords:** chronic stress, cPTSD, glutamate, striatum

## Abstract

Variations in anxiety-related behavior are associated with individual allostatic set-points in chronically stressed rats. Actively offensive rats with the externalizing indicators of sniffling and climbing the stimulus and material tearing during 10 days of predator scent stress had reduced plasma corticosterone, increased striatal glutamate metabolites, and increased adrenal 11-dehydrocorticosterone content compared to passively defensive rats with the internalizing indicators of freezing and grooming, as well as to controls without any behavioral changes. These findings suggest that rats that display active offensive activity in response to stress develop anxiety associated with decreased allostatic set-points and increased resistance to stress.

## 1. Introduction

In the fight for survival, stress-related psychobiological responses are activated together with calming mechanisms [1,2]. Some authors have referred to these adaptive processes as “active (ACS) or passive (PCS) coping styles” [1], while others have described them as active offensive (AOR) or passive defensive responses (PDR). Using a description related to the allostasis paradigm, one may also refer, correspondingly, to the allostatic flight/fight/active (AFR) or allostatic freezing/passive (APR) responses [3]. The AFR involves offensive and proactive investigatory reactions toward environmental threats or stressors, a more aggressive phenotype, and a less pronounced neural and physiological response to stress than the APR. While some researchers have focused on the biological basis of these styles, including hormonal changes, neural remodeling, and gene methylation processes [4,5,6], others have pointed out the importance of individual differences in these responses [7,8,9]. The exact underlying mechanisms of these different psychobiological processes, including genetic and epigenetic changes, are unclear [10,11,12,13,14,15].

Circulating predator scent stress (PSS) glucocorticoid (GC) levels may vary with the different behavioral responses to stress. Thus, in a rat model of chronic PSS, lower plasma GCs were associated with active rather than passive responder animals [1,3]. Similarly, in other animal models of active and passive biobehavioral responses leading to calming down on stress, different allostatic set-points were observed with high and low circulating GC levels, respectively [5,16]. Furthermore, in the forced swim test, rats given subcutaneous injections of corticosterone (CORT) displayed more passive responding and depression-like behaviors than controls [17]. Taken together, these data suggest that elevated circulating CORT levels may play a causal role in promoting passive rather than active styles of responding.

The amygdala, prefrontal cortex (PFC), and ventral striatum are consistent components of the stress response, including synaptic interactions between one of the most important neurotransmitters, i.e., glutamate (Glu), and the circulating levels of CORT [18,19,20,21,22]. While most studies have focused on the circuits between the amygdala and PFC, they have neglected the modulatory role of striatum in the stress response, even when GCs administered directly into the dorsal striatum lead to enhanced consolidation of inhibited avoidance memory in cued water-maze training [23,24,25,26]. Finally, the striatum was linked to individual differences in the organism’s response to stress, with “active coping” associated with striatal activation, as indicated by a heightened glucose uptake in this region during stress [27]. Aside from striatal changes, peculiarities in CORT reduction pathways may contribute to variations in the stress response. The first pathway of CORT reduction reflects limbic–hypothalamic–pituitary–adrenal (LHPA) axis suppression, which is in concordance with reduction of glucose uptake in the hypothalamus of AFR rats [27].

The second pathway may be linked to tissue metabolism of GCs by the 11β hydroxysteroid dehydrogenases (11βHSD) type 1 (11βHSD1) and type 2 (11βHSD2), which are involved in the interconversion between active and inactive forms of GCs, depending on the tissue. Thus, the active form of CORT is metabolized by 11βHSD2 to the inactive form 11-dehydrocorticosterone, while the reverse reaction from an inactive to an active form is catalyzed by the 11βHSD type 1 enzyme [28,29]. The adrenal cortices express both 11βHSD1 and 11βHSD2, which may be involved in not only activating, but also inactivating, calming reactions of the stress response. This suggests that the latter may also take place at the level of the end-organ of the LHPA axis, possibly via a regulated decrease of active GC secretion.

In post-traumatic stress disorder (PTSD), traumatic memories activate the striatum and inactivate the hippocampus, leading to a shift from hippocampal to striatal memory [30,31]. Indeed, functional magnetic resonance imaging (fMRI) studies in PTSD have demonstrated hippocampal–striatal hyperconnectivity, with lack of adaptability and decreased inter-regulation between the two regions [32]. Interestingly, reduced striatum reactivity was associated with increased vulnerability to stress in individuals exposed to early-life stress [33,34]. The effects of the latter on behavioral and striatal development have been well described and include both externalizing and internalizing symptoms [35,36]. Antidepressant-treatment-dependent increases in positive affect in depressed patients can be explained by an increase in sustained nucleus accumbens activity, while reductions of positive affect in this disorder may result in part from loss of the ability to sustain nucleus accumbens activity and connectivity with the fronto-striatal region over time [36]. Adolescents with high aggressiveness exhibited striatal activation during both reward and nonreward phases, whereas healthy controls exhibited striatal activation only during reward, shifting to anterior cingulate activation during nonreward [37].

The genotype/phenotype/endophenotype of individual differences in the psychobiological stress response is of importance and may shed light on the phenomenon of insufficient effectiveness of available therapies for depression and PTSD. According to Taghzouti et al., the antidepressant fluoxetine was effective in only a subgroup of low-stress- but not high-stress-responder rats [38]. These results were confirmed and extended with other drugs, such as desipramine [39] and citalopram or reboxetine [40], which revealed distinct effects of drugs depending on the interindividual differences of allostatic set-points of stress. In light of the above considerations, we examined whether active and passive responses to chronic predator scent test were associated with different allostatic set-points by measuring plasma CORT, striatum Glu metabolites, and adrenal gland CORT and 11-dehydrocorticosterone content. We predicted LHPA axis alterations associated with increased striatal and decreased adrenocortical activity in chronically stressed AFR rats compared to APR or control rats.

## 2. Results

### 2.1. Behavior on Exposure to Stimuli (Kruskal–Wallis One-Way/Bonferroni Post Hoc Tests)

Following exposure to the predator odor stimulus, the rat phenotypic behavioral pattern was classified into one of two groups: allostatic flight/fight response (AFR) and allostatic freezing/passive response (APR). AFR phenotypic behavioral pattern was established in 45% of the animals, and APR in the remaining 55%. In general, climbing, sniffing, and tearing in AFR rats and freezing and grooming in APR rats prevailed in the direct response behavior to stress stimuli, while control rats within their cages demonstrated neutral behavioral patterns. Significant differences between AFR and APR rats were observed in the summarized number of freezing (H_2,26_ = 15.75; *p* < 0.001), grooming (H_2,26_ = 8.85; *p* = 0.012), sniffing (H_2,26_ = 18.88; *p* < 0.001), and climbing (H_2,26_ = 15.17; *p* < 0.001) acts, as well as in the tearing the protective material of the Petri dishes (H_2,26_ = 15.47; *p* < 0.001) acts (Figure 1).

After 10 days of PSS induction, hetero-chronic changes were revealed in AFR rats, which manifested a rise in climbing (with peaks on days 1–3 and 6–7; Figure 1d) and sniffing acts (with peaks on days 1–5 and 9–10; Figure 1c) and episodes of tearing/aggressive behavior (day 1–6; Figure 1e) in comparison to APR rats. AFR rats displayed decreased frequency of freezing on days 2 and 5–6 in comparison to APR rats (Figure 1a), while the latter demonstrated higher numbers of grooming behavior acts on days 7 and 9 in comparison to AFR animals (Figure 1b). In AFR rats, an upward trend in the incidence of grooming was determined (the number of grooming acts at the final stages of the experiment exceeded the ones of the initial stages). Thus, in the dynamics of 10-day PSS, only the aggressiveness was characterized by the presence of traits in AFR rats, with complete absences in APR rats. Other characteristics were revealed in both phenotypes, although with different intensities. This is why we identified the dominant behavioral patterns in the two phenotypes of rats, evaluating the differences in the summarized numbers of behavioral acts during the 10 days of PSS exposure.

### 2.2. AFR and APR Rat Behavior in Elevated Plus Maze

A significant influence of the behavioral phenotype in response to PSS on the cumulative number of entries (F_2,26_ = 12.84; *p* < 0.001), exploring (F_2,26_ = 14.3; *p* < 0.001), and time spent in the open (OA) (F_2,26_ = 21.04; *p* < 0.0001) and closed arms (CA) (F_2,26_ = 23.04; *p* < 0.0001) was revealed (Table 1). According to the results, PSS led to a rise in OA entries (*p* = 0.0025 vs. control; *p* = 0.001 vs. APR), OA exploring, and time spent in OA (*p* = 0.00007 vs. control; *p* = 0.0001 vs. APR) and CA (*p* = 0.0001 vs. control; *p* < 0.0001 vs. APR) in the AFR compared to APR and unstressed control rats. Overall, the different behavioral phenotypes in response to PSS exposures were characterized by the distinctions in animal anxiety levels on day 14 post PSS cessation: diminished anxiety levels were observed only in AFR rats.

### 2.3. Biochemical Differences in Our Classified Behavioral Subtypes

Magnetic resonance spectroscopy (MRS) analysis using a one-way ANOVA (Figure 2) revealed significant differences in summarized Glu+glutamine (Gln) metabolites in striatum (F_2,26_ = 6.97; *p* < 0.001), with higher Glu+Gln levels in AFR (24.94 ± 9.51%, *n* = 9) than in APR rats (17.02 ± 5.19%, *n* = 12; *p* < 0.005) and when compared with control rats (16.37 ± 3.35%; *p* < 0.005), while there were no differences in Glu+Gln levels in APR rats in comparison to control animals.

Even 18 days after PSS induction (Figure 3a), differences in plasma CORT levels were significant (F_2,26_ = 6.71; *p* < 0.01): hormone levels in AFR rats (16.61 ± 6.59 ng/mL, *n* = 9) were lower than in APR (58.63 ± 23.54 ng/mL, *n* = 12; *p* = 0.007) and control animals (41.18 ± 17.23 ng/mL, *n* = 8; *p* = 0.049), while no differences were revealed between APR and control animals.

The adrenal concentrations of CORT and 11-dehydrocorticosterone (as an inactivated derivative of CORT) were examined to evaluate the balance between GC production and inactivation in AFR and APR rats 18 days post PSS cessation. The analysis, conducted via a one-way ANOVA, did not reveal any significant effects of the behavioral response to PSS on CORT (F_2,26_ = 0.74; *p* = 0.52); however, 11-dehydrocorticosterone levels (Figure 3b) were different (F_2,26_ = 3.7; *p* = 0.041) between either group of stressed rats (AFR and APR) and controls: AFR rats (0.85 ± 0.13 ng/mg, *n* = 9) were characterized by higher 11-dehydrocorticosterone levels than APR (0.66 ± 0.049 ng/mg, *n* = 12; *p* = 0.0006) and control (0.69 ± 0.11 ng/mg, *n* = 8; *p* = 0.019) rats, while the 11-dehydrocorticosterone level reduction in APR was similar to that of the control (*p* = 0.25) rats. Moreover, the 11-dehydrocorticosterone/CORT ratio was different (F_2,26_ = 3.7; *p* = 0.041) between both groups of stressed rats (AFR and APR). AFR rats (35.04 ± 11.25 *n* = 9) were characterized by lower CORT/11-dehydrocorticosterone ratio than APR (55.16 ± 16.12 *n* = 12; *p* = 0.0006) and control rats (52.33 ± 18.25; *n* = 8; *p* = 0.033), while the 11-dehydrocorticosterone/CORT ratio in APR rats was similar to that of the control (*p* = 0.25).

## 3. Discussion

The results obtained in this study allowed us to propose a link between “coping style” and related behavioral patterns in PSS exposure. Thus, animals with an offensive response to chronic predator stress (AFR) not only exhibited the lowest anxiety levels in response to the stressor [3], but also had higher striatal glutamate metabolite concentrations, as well as lower basal plasma CORT, and higher 11-dehydrocorticosterone in the adrenal glands than APR rats and controls. This observation is consistent with our previous data, where the anxiety extinction after stress exposure, accompanied by higher lactate (lac) and lower Glu levels in the amygdala, another main driver of the anxiety response [18,19,20,21,22,41] as observed in AFR animals [3]. Taken together, our results suggest a novel mechanism of transient hypocorticosteronemia in tissues, with corticosterone being suppressed in situ in the adrenal glands. It is important that this is reversible and that the pool of the synthesized corticosterone could be restored rapidly (Figure 4).

Granted that active coping style pertains to the flight/fight response, while passive coping style corresponds to freezing/passive response [1,5,16], it makes sense that high CORT levels in APR animals are associated with a passive coping strategy. This is in agreement with data in primates and rodents, in which high basal and stress-induced cortisol levels have been associated with increased freezing [42,43]. Similarly, preventing CORT release in newborn rats by removing the adrenals leads to impaired freezing, which can be restored by glucocorticoid administration [44].

According to Schwabe et al., psychobiological mechanisms of stress-resilience are associated with an active response to stress stimuli and a bias toward the use of stimulus–response (S–R) learning; the striatum is a key player in the implementation of S–R learning, and this was indirectly confirmed by data from striatal NMR spectroscopy [2]. Our method, as applied here, has revealed an increase in striatal Glu+Gln in AFR compared to APR rats. The elevation of the major excitatory neurotransmitter Glu in AFR animals reflects activation of the striatum, which corresponds to S–R learning in rats with an active coping style [2,25,27]. Interestingly, the striatum showed no changes in Glu+Gln during early stress, but rather, a significant increase was observed in the late period, relative to changes in neutral runs [32]. An opposite dynamic response to stress was seen in other regions, including the hippocampus, with increases in neural activity in the early period, followed by reduced activity later; these changes are suggestive of an adaptive or habituation network part of the adaptive response to stress [32]. The observed increase in striatal activity may buffer against the effect of negative experiences on the development of post-traumatic manifestations [45]. This would explain the low anxiety levels we observed in AFR rats.

Elevated levels of the inactive GC 11-dehydrocorticosterone were detected in the adrenal glands of AFR rats. Taking into account that the rat adrenal glands have much higher 11βHSD2 than 11βHSD1 activity [46], we could not rule out a decrease of CORT release by 11βHSD2 activation preventing its secretion. The complexity of 11βHSD action has been highlighted by studies of 11βHSD1 knockout mice, which display compensatory adrenal hyperplasia and increased basal levels of corticosterone, despite the presumed absence of hepatic 11-dehydrocorticosterone to corticosterone conversion [47]. In addition, 11βHSD2 provides protection of the target tissues and modulates circulating levels of CORT, as shown in birds [48]. Moreover, peripheral antagonism of the 11βHSD system has a greater impact on circulating glucocorticoid levels than central control during the stress response, and this system, in turn, could be influenced by ACTH, which caused a 5–10 fold increase in 11ßHSD2 mRNA in primary cultures of rat adrenocortical cells [49]. The unique advantage of such local regulation is that the adrenal cortex could rapidly and reversibly modify the secreted ratio of the inactive to the active form of CORT. A rapid transition of this ratio might serve as an advantage of AFR over APR animals by preventing negative effects associated with a long-term decrease in GC levels. As the adrenal cortices express both 11βHSD1 and 11βHSD2, we postulate that both activating and inactivating reactions may take place, representing a previously unsuspected rapid regulatory mechanism of the stress response.

## 4. Materials and Methods

Experiments were performed on 29 genetically similar white male Sprague Dawley rats, each weighing 240–260 g (age of 8–9 weeks), obtained from the specific-pathogen-free (SPF) vivarium of the Institute of Cytology and Genetics SB RAS (Novosibirsk, Russian Federation). Rats were housed in sibling pairs in standard ventilated cages (IVC BlueLine, Tecniplast, Italy). Water and granulated forage (Sniff, Soest, Germany) were given ad libitum. Animals were kept in a 14 h light (2 a.m. to 4 p.m.) and 10 h dark (4 p.m. to 2 a.m.) cycle, temperature (22–24 °C), and relative humidity (40–50%) controlled environment. The behavioral task was always initiated at the start of the dark cycle, when rodents are most active.

All animal experiments conformed to the requirements of the Council for International Organizations of Medical Sciences (CIOMS) and the International Council for Laboratory Animal Science (ICLAS), as described in “International Guiding Principles for Biomedical Research Involving Animals” (2012). The handling of all animals was identical. The study protocol was approved by the Committee for Bioethics and Humane Treatment of Laboratory Animals at South Ural State University, Russia (Project 0425-2018-0011 from 17 May 2018, protocol number 27/521).

### 4.1. Experimental Procedure

Following exposure to the predator odor stimulus, the rat phenotypic behavioral pattern was classified into one of two groups: allostatically active flight/fight response (AFR), i.e., rats exhibiting a “stimulus–response” behavior pattern, and allostatically freezing/passive response (APR) groups [3,50,51].

For the PSS protocol, rats were exposed to cat urine scent in a Petri dish with litter for 10 min daily for 10 days (21 rats were submitted to stress exposure; 8 control rats were exposed to a neutral scent). Repeated exposure to the PSS may be a more accurate model of human PTSD than the single acute exposure approach, granted that it minimizes the effect of confounding factors, such as the concentration of pheromones in each individual urine scent exposure [3,52]. All procedures were performed between 1:00 and 2:00 p.m. During the scent exposure protocol, stress-related behavior was captured daily via web-camera. Behavioral evaluation was conducted via 3D animal tracking system “EthoStudio” [53]. The evaluator of the behavior had not previously worked with any rats in our groups. Recorded variables included the time spent in open and closed arms of the maze and the number of entries into the open and closed arms.

The timeline for modeling PSS, evaluating stress-related behavior and anxiety, and measuring of metabolites (CORT, Glu+Gln and 11-dehydrocorticosterone) in plasma, brain, and adrenal glands, respectively, was as follows:Days 1–10: PSS;Days 11–22: rest;Day 23: elevated plus maze test;Day 27: striatum metabolite measurement by MRS;Day 28: euthanasia, harvesting of blood and organs.

### 4.2. Behavioral Activity

Video recordings of PSS sessions were made in the home cages. The presence or absence of behavioral responses was recorded daily. The frequencies of freezing, grooming, sniffing of stimuli, climbing on stimuli, and tearing of protective cover of stimuli were used for classification of rats as AFR and APR. The presence of the response in one session was marked with “1”, while the lack of a response was marked as “0”. Apart from registration of the daily changes of the observed behavioral responses, we also summarized the frequencies of these behavioral responses over 10 days. The predator stress outcome was evaluated using the elevated plus maze test, using the standard elevated plus maze (EPM) test apparatus TS0502-R3 (OpenScience, Russia) [54,55]. Variables recorded included time spent in open and closed arms of the maze and the number of entries into the open and closed arms. While Table 1 demonstrates the measurements of the maze, Figure 1 reflects the dynamics of rat behavior in home cages during the 10 days of PSS. 

### 4.3. Magnetic Resonance Spectroscopy (MRS)

Rat striatum neurometabolites (Figure 2) were measured on a horizontal tomograph with a magnetic field of 11.7 tesla (Bruker, Biospec 117/16 USR, Germany). The rats were anesthetized with gas (isoflurane; Baxter Healthcare Corp., Deerfield, IL, USA) using a Univentor 400 Anesthesia Unit (Univentor, Zejtun, Malta). The tomographic table contained a water circuit that maintained a surface temperature of 30 °C to preserve animal body temperature during the test. A pneumatic respiration sensor (SA Instruments, Stony Brook, NY), placed under the lower body, controlled the depth of anesthesia. Proton spectra of the rat striatum were recorded with transmitter volume (T11232V3) and rat brain receiver surface (T11425V3) using 1 Hz radiofrequency coils (Bruker, Ettlingen, Germany). High-resolution T2-weighted images of the rat brain in three (axial, sagittal, and coronal) dimensions (section thickness, 0.5 mm; field of view, 2.5 × 2.5 cm for axial and 3.0 × 3.0 cm for sagittal and coronal sections; matrix of 256 × 256 dots) were recorded by rapid acquisition with relaxation enhancement (TurboRARE), with the pulse sequence parameters TE = 11 ms and TR = 2.5 s for correct positioning of the spectroscopic voxels. Voxel dimension was 3.0 × 3.0 × 3.0 mm for the striatum. Voxel was manually placed according to a structural T2-weighted MRI image (Figure 5). All proton spectra were recorded by spatially localized single-voxel stimulated echo acquisition mode (STEAM) spectroscopy, with the following pulse sequence parameters: TE = 3 ms, TR = 5 s, and 120 accumulations. Uniformity of the magnetic field was tuned within the selected voxel using FastMap [56] before each spectroscopic recording. The water signal was inhibited with a variable pulse power and optimized relaxation delays (VAPOR) sequence [57].

### 4.4. Processing of ^1^H Spectra

The experimental ^1^H magnetic resonance spectra were processed, and the quantitative composition of metabolites was determined with a custom-made program similar to that of the LC Model software package [58,59]. The baseline correction was conducted automatically by the program to determine the spectral baseline for fitting of the spectrum obtained by ^1^H MRS. The process of fitting was presented on the real-time plot, and the fitted data were stored in numerical form.

### 4.5. Hormonal Measurements

Between 11:00 a.m. and 1:00 p.m. on experimental day 28, rats were sacrificed by decapitation, and blood samples were collected in tubes with heparin. Blood samples (Figure 3) were then centrifuged at 3000 rpm at 4 °C for 15 min. Plasma samples were aliquoted and stored in a −80 °C freezer until use. After thawing, plasma CORT concentrations were measured by ELISA (Cusabio ELISA Kit, Texas, USA) as per manufacturer’s instructions. The assay sensitivity was 0.25 ng/mL, and the intra- and interassay coefficients of variation were both <5%.

High-performance liquid chromatography (HPLC) using the micro-column liquid chromatograph Milikhrom-1 (NPO “Nauchpribor”, Orel, Russia) for the evaluation of corticosteroids (CORT and 11-dehydrocorticosterone) in rat adrenal glands (Figure 4) was carried out on experimental day 28. The entire procedure of adrenal CORT extraction, measurements, and validation was previously described in detail [60,61]. The adrenal glands were weighed, transferred into a glass homogenizer, placed into an ice bath, and thoroughly homogenized in 1 mL of cold acetone. The homogenates were transferred to plastic tubes, and the samples were centrifuged at 2000 g at 4 °C for 15 min. The supernatants were then poured into a plastic tube and evaporated in a nitrogen flow at 40 °C. The residues were dissolved in 24 µL of a 65% solution of CH3OH in water. The injection volumes were 8 μL. Determinations of corticosteroid hormones were carried out using micro-column high-performance liquid chromatography (HPLC) [62]. Chromatographic conditions were as follows: steel column 2 × 62 mm in size, packed with Silasorb C18 SPH 8 (5 µm) as sorbent, gradient elution with acetonitrile in water from 30 to 55% (*v*/*v*), eluting rate of 100 µL/min, and UV detection at 240 and 260 nm. The wavelengths were chosen because of the absorption spectrum of steroid hormones: 240 nm corresponds to the maximal absorption of the majority of corticosteroid hormones, while 260 nm corresponds to half of the maximal absorption. Chromatographic information was processed with the help of CHROM software (EcoNova, Institute of Chromatography, Novosibirsk, Russia).

Hormone identification was performed by comparing retention times and spectral ratios of endogenous corticosteroids and synthetic standards. The amounts of the corticosteroids were determined in nanograms per mg of tissue (ng/mg) using calibration curves plotted individually for each hormone under investigation. Chromatographic separation of a mixture of standards (a) (C = 10 ng/µL, 5 μL was taken for the analysis) and adrenal extracts (b) is shown in Figure 6. 

### 4.6. Data Analyses

Data were analyzed with SPSS 24, STATISTICA 10.0 and MS Excel software. Quantitative data were presented as mean ± SEM. After having shown normal distribution by the Shapiro–Wilk test, a one-way ANOVA with Fisher’s LSD post hoc tests and the Kruskal–Wallis with Bonferroni corrected post hoc tests were used to compare all outcome measures between two groups (e.g., control vs. AFR, control vs. APR, AFR vs. APR). *p* < 0.05 was considered significant.

## 5. Conclusions

The LHPA axis coordinates behavioral and physical responses to chronic stressors, with the striatum acting as a player in the implementation of the active offensive behavior. GCs modulate the activity of the striatum. In our study, we observed activation of the striatum, an important component of an active behavioral strategy that was accompanied by decreased GC levels. This decrease was seen in a relatively distant period from the timing of the stressor. Other investigators described activation of the striatum at earlier time intervals relative to the timing of the stressor, which was probably associated with elevated GC levels. Lower CORT levels in blood might be related to decreased activity of the HPA axis and/or increased adrenal conversion to 11-dehydrocorticosterone. In the latter case, we demonstrated a transient hypocorticosteronemia, which might quickly level out or turn into hypercorticosteronemia. Our data might have some clinical implications suggesting that individuals with more externalizing behaviors (more active offensive response to chronic stress), have less anxiety and lower basal CORT levels compared to individuals with more internalizing behaviors (freezing/passive response to chronic stress), indicating a lower allostatic set-point with higher resistance to stress in externalizing individuals.

## 6. Limitations of the Study

The main limitation of this study was that we did not investigate the activity and expression of 11βHSD1 and -2 in peripheral tissues beyond the adrenal glands. Further studies will examine this issue to establish adrenal and non-adrenal factors that might regulate CORT metabolism. We should also emphasize that CORT levels and patterns discussed here might not correspond to those that could be observed after the initial exposure. Thus, according to Dopfel et al., in low-freezing (i.e., active) animals, the CORT response immediately after stress exposure was heightened and prolonged and had a delayed return to baseline, a pattern of CORT response that has been suggested as a marker of maladaptation to stress in individuals who develop PTSD [9]. With time, this pattern of CORT response may change through several processes, including changes in adrenal enzymatic activity, as shown in this study.

## Figures and Tables

**Figure 1 ijms-21-07440-f001:**
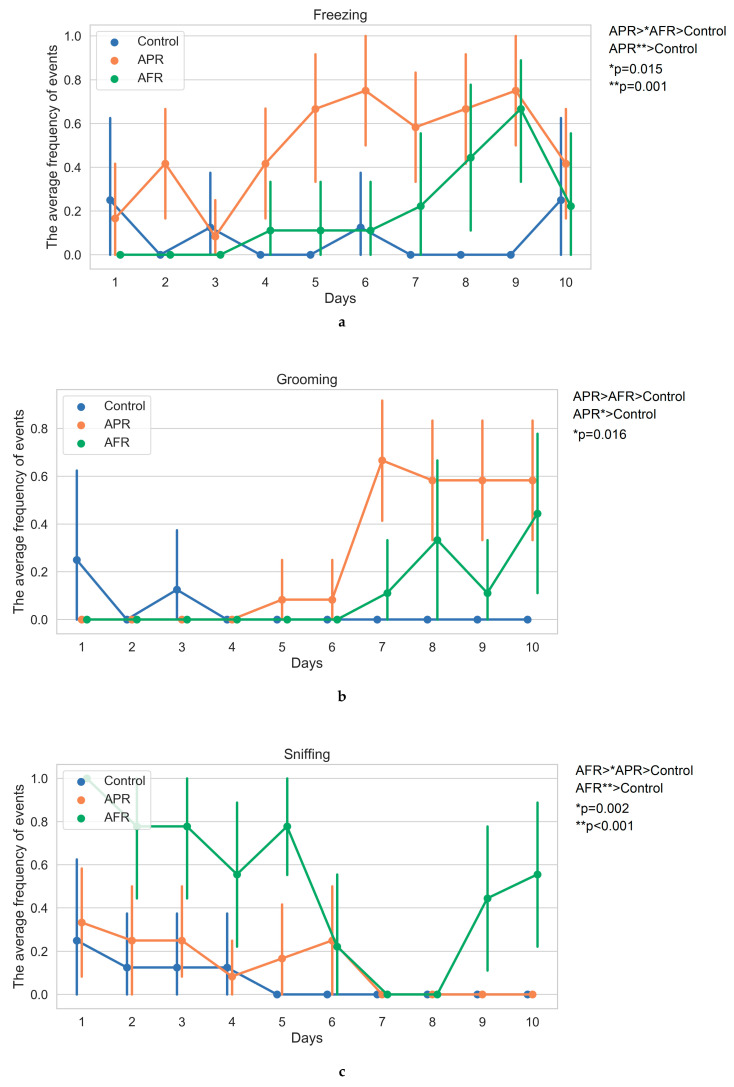
Behavior patterns of allostatic flight/fight/active (AFR) or allostatic freezing/passive (APR) response rats. Legend: Bonferroni’s calculations for M ± SE of (**a**) the frequency (per rat) of freezing behavior acts, (**b**) the frequency (per rat) of grooming behavior acts, (**c**) The frequency (per rat) of sniffing acts, (**d**) the frequency (per rat) of climbing acts, (**e**) the frequency (per rat) of attempts to tear the protective material of the Petri dishes.

**Figure 2 ijms-21-07440-f002:**
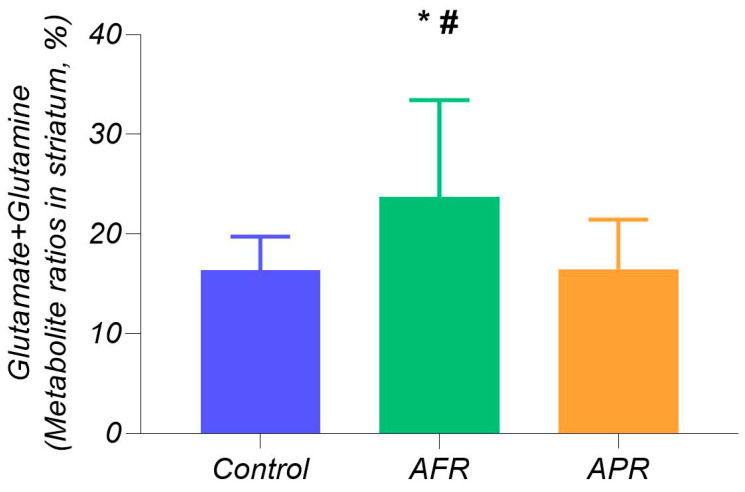
Long-term consequences of predator scent stress (PSS) in rat striatum. Legend: Total metabolite (%) glutamate+glutamine concentrations. * *p* < 0.05 in comparison with APR (*n* = 12); # *p* < 0.05 AFR (*n* = 9) in comparison with control.

**Figure 3 ijms-21-07440-f003:**
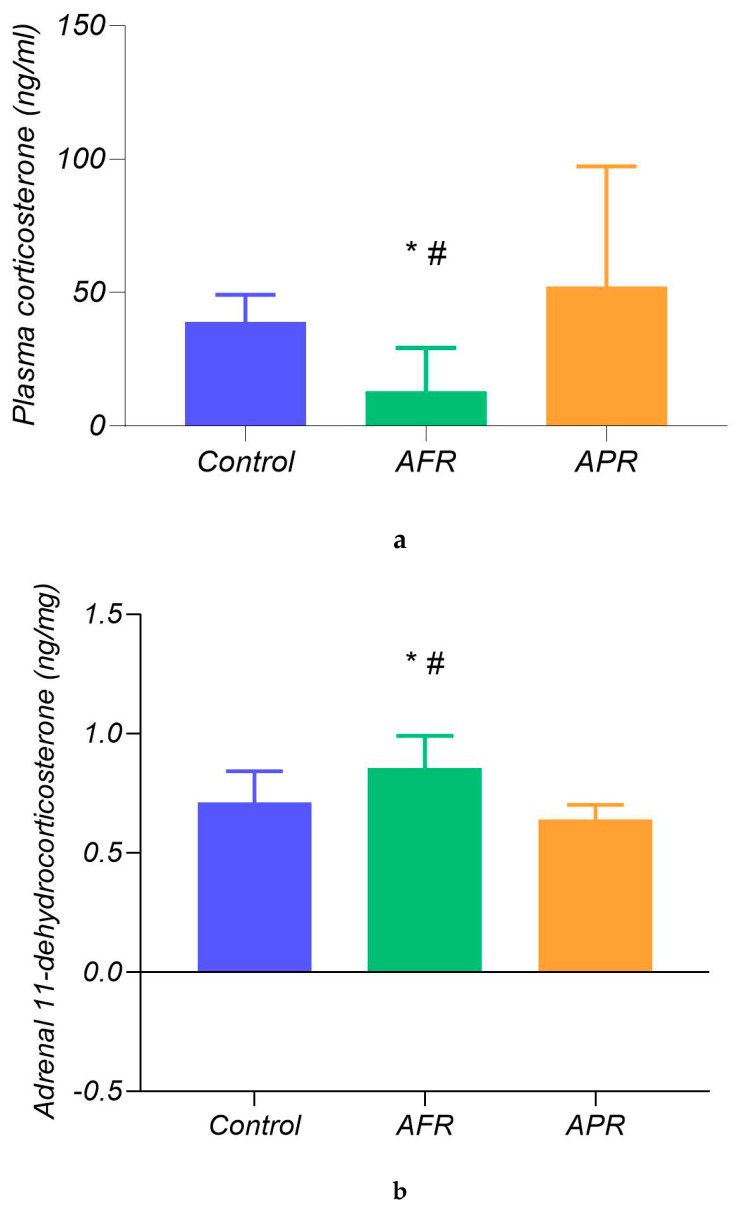
Comparison of peripheral corticosteroids 18 days after PSS. Legend: (**a**) Plasma CORT levels (ng/mL) and (**b**) adrenal 11-dehydrocorticosterone levels (ng/mg of tissue). # *p* < 0.05 AFR (*n* = 9) in comparison to APR (*n* = 12).); * *p* < 0.05 in comparison to control (*n* = 8).

**Figure 4 ijms-21-07440-f004:**
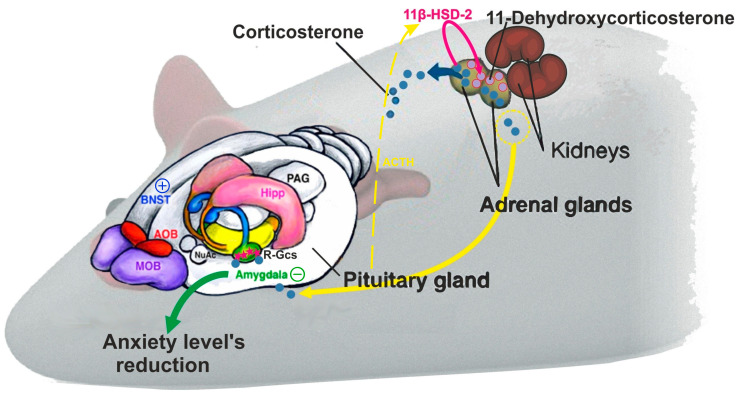
Parallel behavioral and brain biochemical changes in AFR rats. Legend: Hypocorticosteronemia in AFR rats develops as a result of corticosterone inactivation by 11βHSD2 in the adrenal glands. Low corticosterone levels contribute to inhibition of amygdala activity, which manifests as a decrease in glutamate (Glu)/glutamine (Gln) ratio and an increase in lactate (Lac) in this brain structure [3]. A parallel increase in the excitatory neurotransmitter Glu+Gln in the striatum possibly indicates activation of this brain structure. Inhibition of the amygdala and activation of the striatum possibly lead to a decrease in the anxiety of AFR rats. The observed decrease in corticosterone is most likely transient and rapidly restorable. The red stars indicated “R-Gcs” (Gluccocorticoid receptors of the amygdala).

**Figure 5 ijms-21-07440-f005:**
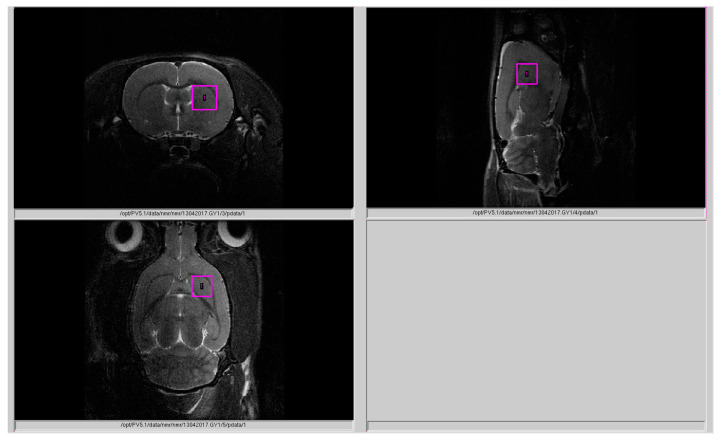
MRS measurements in the striatal region. The labeled pink area indicated the: “Voxel position during 1H MRS of the striatum”.

**Figure 6 ijms-21-07440-f006:**
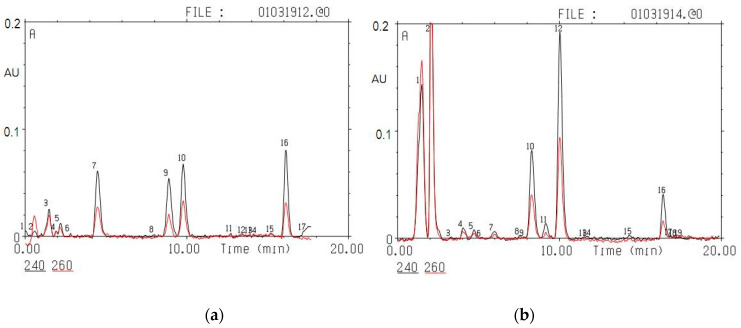
Chromatographic separation of a mixture of standards (**a**): 7-aldosterone, 9 – 11-dehydrocorticosterone, 10-corticosterone, and 16 – 11-desoxycorticosterone; and adrenal extracts (**b**): 5-aldosterone, 11 – 11-dehydrocorticosterone, 12-corticosterone, and 16 – 11-desoxycorticosterone. The black line corresponds to absorption at 240 nm, the red line to absorption at 260 nm. The y-axis is absorption (A) in absorption units (AU); the x-axis is time in min.

**Table 1 ijms-21-07440-t001:** Behavioral performance of AFR and APR rats in elevated plus maze test.

	Control	APR	AFR
Central square time	0.10 ± 0.02	0.12 ± 0.02	0.19 ± 0.01 *^,#^
% Closed arms time	0.89 ± 0.03	0.8 ± 0.03	0.62 ± 0.03 *^,#^
% Open arms time	0.11 ± 0.01	0.08 ± 0.01	0.19 ± 0.03 *^,#^
Entries into open arms	1.32 ± 0.24	1.15 ± 0.024	4.42 ± 0.45 *^,#^
Entries into closed arms	4.55 ± 0.78	8.68 ± 1.25	5.43 ± 0.35

Legend: Data presented using M ± SD; * *p* < 0.05 in comparison with control (*n* = 8); ^#^
*p* < 0.05 AFR (*n* = 9) in comparison with APR (*n* = 12).

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
