# Peer review of "Offensive Behavior, Striatal Glutamate Metabolites, and Limbic–Hypothalamic–Pituitary–Adrenal Responses to Stress in Chronic Anxiety"

_ijms, 2020, doi:10.3390/ijms21207440_

Round 1

Reviewer 1 Report

The study investigated whether active (AFR) and passive (APR) responses to chronic PSS are leading to different allostatic set points measured within plasma (CORT), striatum (glutamate) and the adrenal glands (11-dehydrocorticosterone, CORT). The results are interesting and potentially important; however, there are some issues that need to be addressed.

Introduction is too long; I suggest moving large parts of the text to Section Discussion.

Results:

Figure 1 is not clear. What is on the y axis? The asterisks should be shown on each significant point in the graphs. The graphs should be better arranged, so it would be clear what each graph is presenting.

Figure 2-desripriton of each picture in panel is missing

In each Figure description the authors should write the statistical tests used.

Could active (AFR) and passive (APR) rats be predicted based on baseline CORT or behaviour? Have authors measured baseline levels?

Materials and methods:

The authors wrote that experiments were performed on 29 white genetically similar male rat. What rat strain did they use? How old were the rats?

Why have authors conducted behavioural tests (elevated plus maze) only 12 days after PSS and other tests even later (18 days)? What is happening write after PSS and during 10 days after PSS?

Data Analyses: The authors wrote that they used One-way ANOVA with Fisher-LSD post hoc tests and the Kruskal Wallis with Bonferroni post hoc tests to compare all outcome measures between two groups. These parametric and nonparametric tests are used when comparing three or more groups. How have authors checked the normality of data?

Overall, the findings presented are very similar to the results from the previous study of the same authors (Ullmann, E.; Perry, S.W.; Licinio, J.; Wong, M.L.; Dremencov, E.; Zavjalov, E.L.; Shevelev, O.B.; Khotskin, N.V.; Koncevaya, G.V.; Khotshkina, A.S.; et al. From allostatic load to allostatic state--An endogenous sympathetic strategy to deal with chronic anxiety and stress? Frontiers in Behavioral Neuroscience 2019 doi10.3389/fnbeh.2019.00047). Both studies used the same model of chronic PSS and same methods, although the previous paper investigated hippocampus and amygdala and the present study addressed changes in the striatum. Therefore, I suggest that authors make one additional figure summarizing previous and present findings and proposing mechanisms involved in differential responses to PSS between active AFR and APR rats.

There are some language and writing errors, which need to be corrected.

Author Response

  • Introduction is too long; I suggest moving large parts of the text to Section Discussion:
  • We are thankful for this suggestion. We have made the appropriate changes in the Introduction and Discussion sections.
  • “Figure 1 is not clear. What is on the y axis? The asterisks should be shown on each significant point in the graphs. The graphs should be better arranged, so it would be clear what each graph is presenting.”; “Figure 2-desripriton of each picture in panel is missing”; “In each Figure description the authors should write the statistical tests used.”
  • Thank you very much for this useful piece of advise. Please note that we have replaced all pictures with the appropriate changes in each figure.
  • Could active (AFR) and passive (APR) rats be predicted based on baseline CORT or behaviour? Have authors measured baseline levels?”
  • We appreciate this very useful piece of advice, however, in this protocol, we did not measure basal corticosterone levels in parallel to behavioral activity, because the blood collection procedure itself is invasive and could result in aberrations of the behavioral responses observed. Our experience from previous research has convinced us of the validity of these concerns. In particular, we have observed that the invasive character of the procedure affects the behavioral responses of stressed rats negatively (appearance of freezing in most rats in the first three days of the experiment, not only in the stressed animals but also in the controls).

We also elected to assess the basal level of anxiety, because testing at the entry can affect the indicators of anxiety at the exit, as there is loss of the animal's sense of novelty in response to the behavioral stressor in the cruciform maze.

However, we understand that the stressed animals differ in their behavioral and neuroendocrine status. Hence, to mitigate the effect of these differences we carefully randomized the animals. We concluded that with this technique, we were able to significantly reduce the contribution of the initial state to the response of the animals during and after the predator stress.

  • “The authors wrote that experiments were performed on 29 white genetically similar male rat. What rat strain did they use? How old were the rats?”
  • Thank you for this clarification. We have now included this information in the Methods section as follows: “Experiments were performed on 29 genetically similar white male Sprague-Dawley rats, each weighing 240–260 g (age of 8-9 weeks)…” (page 9, line 273).
  • “Why have authors conducted behavioural tests (elevated plus maze) only 12 days after PSS and other tests even later (18 days)? What is happening write after PSS and during 10 days after PSS?”
  • Thank you very much for your advise on this issue. We used an experimental model of PTSD. The latter develops manifestations long after the end of the stressful event that provoked it. We considered 12 days as the starting point of the follow-up. Other researchers studied behavior at more distant time-points [Zoladz PR, Diamond DM. Predator-based animal model of PTSD: Preclinical assessment of traumatic stress at the cognitive, hormonal, pharmacological, cardiovascular and epigenetic levels of analysis. Exp. Neurol. 2016; 284 (Pt B): 211-219]. We have previously demonstrated that anxiety symptoms appear only after the tenth day post predatory stress [Tseilikman, V.E., Lapshin, M.S., Komel’kova, M.V. et al. Dynamics of Changes in GABA and Catecholamine Contents and MAO-A Activity in Experimental Post Traumatic Stress Disorder in Rats. Neurosci Behav Physi 49, 754–758 (2019).]. In the first three days, all rats have reduced anxiety and this is associated with increased concentrations of the inhibitory mediator GABA in the brain.
  • Data Analyses: The authors wrote that they used One-way ANOVA with Fisher-LSD post hoc tests and the Kruskal Wallis with Bonferroni post hoc tests to compare all outcome measures between two groups. These parametric and nonparametric tests are used when comparing three or more groups. How have authors checked the normality of data?”
  • We appreciate this comment and have now included the following information in the Methods section: “After having shown normal distribution by the Shapiro-Wilk test a…” (page 11, line 370).
  1. Overall, the findings presented are very similar to the results from the previous study of the same authors (Ullmann, E.; Perry, S.W.; Licinio, J.; Wong, M.L.; Dremencov, E.; Zavjalov, E.L.; Shevelev, O.B.; Khotskin, N.V.; Koncevaya, G.V.; Khotshkina, A.S.; et al. From allostatic load to allostatic state--An endogenous sympathetic strategy to deal with chronic anxiety and stress? Frontiers in Behavioral Neuroscience 2019 doi10.3389/fnbeh.2019.00047). Both studies used the same model of chronic PSS and same methods, although the previous paper investigated hippocampus and amygdala and the present study addressed changes in the striatum. Therefore, I suggest that authors make one additional figure summarizing previous and present findings and proposing mechanisms involved in differential responses to PSS between active AFR and APR rats.” “There are some language and writing errors, which need to be corrected.”
  • Thank you very much for this helpful advise. These studies indicated similar presence of low glucocorticoid levels and decreased anxiety in the more active rats. However, the rest of this work is significantly different. Our previous article provided data on the hippocampus and amygdala, whereas the striatum analyses are described here. Furthermore, adrenal gland steroidogenesis and metabolism were analyzed to explain the status of hypocorticosteronemia. Possibly, these data suggest a novel mechanism of transient hypocorticosteronemia with suppression at the level of the synthesis of the steroid hormone. It is important that this suppression is reversible and that the pool of the synthesized corticosterone restored rapidly and this is possible and discussed in the manuscript. Based on your suggestions, we have created a new figure and described the mechanism within the Discussion section: “Taken together, our results suggest a novel mechanism of transient hypocorticosteronemia in tissues, with corticosterone being suppressed in situ in the adrenal glands. It is important that this is reversible and that the pool of the synthesized corticosterone could be restored rapidly (Figure 4).” (page 8, line 220)
  •  
  •  

Figure 4: Parallel behavioral and brain biochemical changes in AFR rats.

Legend: Hypocorticosteronemia in AFR rats develops as a result of corticosterone inactivation by 11βHSD2 in the adrenal glands. Low corticosterone levels contribute to inhibition of amygdala activity, which manifests as a decrease in Glu:Gln ratio and an increase in Lac in this brain structure [3]. A parallel increase in the excitatory neurotransmitter Glu+Gln in the striatum possibly indicates activation of this brain structure. Inhibition of the amygdala and activation of the striatum possibly lead to a decrease in the anxiety of AFR rats. The observed decrease in corticosterone is most likely transient and rapidly restorable.

Reviewer 2 Report

Ullman and colleagues submitted an interesting manuscript to the journal.  The article aims to understand the underlying mechanisms of how the brain responds to anxiety and stress.  The work is important, as investigating PTSD related mechanisms are not well understood.  There are a few issues that should be addressed prior to publication.

  • Active/Passive responders are mentioned in one figure and APR/AFR in others. Consistency would help the reader understand the data better.  IF these are different distinctions, that should be clearly stated.  Including the more general description is certainly suitable in the discussion and introduction, but less suitable in results and methods.
  • The data is interesting that corticosterone levels are lower in active responders. Do the authors have another other evidence of changes due to stress presumably via hypercortisolemia are attenuated?  For example, do these animals have differences in adrenal hypertrophy, thymic involution, food intake, or body weight gain.  Other measures would also be suitable to support this view.

Author Response

  • Active/Passive responders are mentioned in one figure and APR/AFR in others. Consistency would help the reader understand the data better.  IF these are different distinctions, that should be clearly stated.  Including the more general description is certainly suitable in the discussion and introduction, but less suitable in results and methods.”
  • Thank you for this important comment. According to point 1 of this letter, we replaced all pictures with the changes suggested in figure 1-3.
  1. The data is interesting that corticosterone levels are lower in active responders. Do the authors have another other evidence of changes due to stress presumably via hypercortisolemia are attenuated?  For example, do these animals have differences in adrenal hypertrophy, thymic involution, food intake, or body weight gain. Other measures would also be suitable to support this view.”
  2.  
  • We appreciate this comment. Indeed, in other unpublished studies performed in Wistar rats, we showed that active rats with low corticosterone had a larger thymus, with increased karyocytes. Inadvertedly, we did not examine this issue in this study. According to point 7 of this letter, we created a new figure and described the mechanism within the discussion section: “Taken together, our results suggest a novel mechanism of transient hypocorticosteronemia in tissues, with corticosterone being suppressed in situ in the adrenal glands. It is important that this is reversible and that the pool of the synthesized corticosterone could be restored rapidly (Figure 4).” (page 8, line 220)
  •  

Reviewer 3 Report

MAJOR COMMENTS

The Introduction is sometimes difficult to follow, in part to due to unclear explanations of background data. For example

- lines 57-59 - are the authors discussing basal or PSS-induced GC levels?

- lines 64-65 - "basal" and "elevated" cort usually refer to separate conditions, but this sentence refers to "the basal elevated circulating CORT" which I find confusing.

- lines 66-69 - the authors cite data showing that circulating cort returns to normal within 60 min after cat fur exposure, but then say this is evidence that "long-lasting elevation of CORT in passive responders is a pre-existing characteristic". Did the cat study compare active and passive responders at all? I don't see the connection here, so perhaps more explanation would be helpful.

It is unclear to me exactly how rats were characterized as APR or AFR. The methods state that the frequencies of various behaviors were used, but not how these numbers were used. Does a "1" value mean the presence of any response? Or each response (freezing, grooming, sniffing, climbing, tearing" received a 0 or 1 value? And how were these values used to categorize individual rats?

How were steroids extracted for HPLC analysis? How was the method validated? Were controls run to demonstrate accurate quantification of tissue steroids? UV detection of steroids can be difficult, and DHC measurements are especially lacking in the literature due to technical difficulties. This makes the present data very interesting, but also means validations are critical. These info could be included in Figure 3, or as supplementary data if the authors prefer.

Figure 4

- how is the DHC quantity normalized? ng/mg of tissue? ng/mg protein?

- it would be helpful to include adrenal CORT and plasma DHC data in the figure, and additionally the DHC/CORT ratio in both tissues. This ratio would be the most important when inferring 11betaHSD2 activity. If the ratio in the plasma and adrenal are the same, it is most likely a reflection of 11betaHSD activities elsewhere in the body (i.e. kidney 11betaHSD2 and liver 11betaHSD1), while if the ratio in adrenal differs from that in the plasma, that would be consistent with an adrenal-intrinsic 11betaHSD enzyme activity.

The manuscript would benefit from english language/grammar editing.

MINOR COMMENTS

Please define all acronyms when they first occur (e.g. LHPA axis on line 81, PTSD on line 91, MRI on line 93, MRS on line 171).

Line 56 - change "Neuroendocrinological" to "Neuroendocrine"

Line 73 - the statement that glutamine is "the most important neurotransmitter" should probably be changed, there are of course multiple neurotransmitters that are all absolutely critical.

Line 88-89: If possible, please cite a reference showing that adrenal 11betaHSD2 expression is capable of dampening cort secretion. Since adrenal Cyp11b1 is orders of magnitude higher than Hsd11b1, which is orders of magnitude higher than Hsd11b2, this does not seem very likely.

Lines 140-145: Were temporal changes in behavior tested statistically? It appears that the results describe changes in the means but have no accompanying analyses to support these conclusions.

What rat strain was used? "Genetically similar" is completely vague - please give a more specific description of what this means. Why were only male rats used for this study?

CIOMS has released much more recent "International guiding principles for biomedical research involving animals", at least as recently as 2012, perhaps this would be a more up to date guide to use.

FIGURES

Figure 1

- Please label the y-axes on the graphs, this would make the figures much easier to read.

- Error bars below 0 suggest that the data are not normally distributed, perhaps median and quartile ranges would be a more appropriate way to graph the variation in the data? Offsetting the different lines from each other might also help the reader to see the error bars more clearly.

- caption perhaps AOR and PDR should be switched to APR and AFR?

Figure 3

- this should be labeled Figure 2

- the y-axis reads "Glutamate+Glutamine" but according to the text I think this should be "Glutamate/Glutamine", is that correct? This is also called Glu+Gln in line 174, and should be changed here too.

- perhaps the control group could be the first bar, to be consistent with the ordering of Table 1

Line 208: Do newborn rats produce cortisone? And are there data showing that 11betaHSD1 is important for this cortisone to promote freezing?

There is no explanation of why the glutamine/glutamate ratio is used - do this authors interpret this as a greater production of glutamate? Greater metabolic breakdown of glutamine? Some combination of both?

Author Response

  • The Introduction is sometimes difficult to follow, in part to due to unclear explanations of background data. For example

- lines 57-59 - are the authors discussing basal or PSS-induced GC levels?

- lines 64-65 - "basal" and "elevated" cort usually refer to separate conditions, but this sentence refers to "the basal elevated circulating CORT" which I find confusing.

- lines 66-69 - the authors cite data showing that circulating cort returns to normal within 60 min after cat fur exposure, but then say this is evidence that "long-lasting elevation of CORT in passive responders is a pre-existing characteristic". Did the cat study compare active and passive responders at all? I don't see the connection here, so perhaps more explanation would be helpful.”

  • In response to this comment and according to the initial suggestions of Reviewer 1, we have reduced the Introduction section and clarified some cumbersome formulations.
  •  
  1. it is unclear to me exactly how rats were characterized as APR or AFR. The methods state that the frequencies of various behaviors were used, but not how these numbers were used. Does a "1" value mean the presence of any response? Or each response (freezing, grooming, sniffing, climbing, tearing" received a 0 or 1 value? And how were these values used to categorize individual rats?”
  • We appreciate this comment. The positive reaction was estimated as 1 (one), and the absence of a reaction as 0 (null). Collectively, the daily results were subsequently subjected to statistical processing.

  1. How were steroids extracted for HPLC analysis? How was the method validated? Were controls run to demonstrate accurate quantification of tissue steroids? UV detection of steroids can be difficult, and DHC measurements are especially lacking in the literature due to technical difficulties. This makes the present data very interesting, but also means validations are critical. These info could be included in Figure 3, or as supplementary data if the authors prefer.”

  • Thank you for this important piece of advise. We have now included the following sentence in the Methods section:
  • “The entire procedure of adrenal CORT extraction, measurements and validation was previously described in detail (Selyatitskaya et al. 2008; Baĭkova et al. 1989).“ (page 11, line 360)
  1. Figure 4

- how is the DHC quantity normalized? ng/mg of tissue? ng/mg protein?

  • Thank you for this important observation. We now clarify the units in the legend: “Figure 3. Comparison of peripheral corticosteroids 18 days after PSS. Legend: 3a) plasma CORT levels (ng/ml) and 3b) adrenal 11- dehydrocorticosterone levels (ng/mg of tissue). *p<0.05 in comparison to control (n=8); #P<0.05, AFR (n=9) in comparison to APR (n=12).”

- it would be helpful to include adrenal CORT and plasma DHC data in the figure, and additionally the DHC/CORT ratio in both tissues. This ratio would be the most important when inferring 11betaHSD2 activity. If the ratio in the plasma and adrenal are the same, it is most likely a reflection of 11betaHSD activities elsewhere in the body (i.e. kidney 11betaHSD2 and liver 11betaHSD1), while if the ratio in adrenal differs from that in the plasma, that would be consistent with an adrenal-intrinsic 11betaHSD enzyme activity.”

-> Thank you very much for this insightful comment. In this protocol, we used plasma exclusively for the enzyme immunoassay of corticosterone. Therefore, we cannot provide this information. In later protocols, we measured DHC in plasma and, therefore, were able to calculate this ratio. Unfortunately, we cannot include such data in this article, granted that here the data are given in Sprague-Dawley, while in later protocols they were obtained in Wistar rats. However, here we can provide data on the activity of 11betaHSD2 in the kidneys and 11betaHSD1 in the liver. In neither case did the activities of these enzymes change. Therefore, here we can assume that traditional markers of tissue metabolism (11betaHSD2 in the kidneys and 11betaHSD1 in the liver) glucocorticoids are not involved in a decrease in the level of corticosterone in the blood.

  1. The manuscript would benefit from english language/grammar editing.”
  • The entire manuscript was edited by a native English speaker.
  1. Please define all acronyms when they first occur (e.g. LHPA axis on line 81, PTSD on line 91, MRI on line 93, MRS on line 171).”
  • We are thankful for these suggestions and have made the appropriate changes accordingly.
  1. Line 56 - change "Neuroendocrinological" to "Neuroendocrine"”
  • Thank you. This was changed as suggested.
  1. Line 73 - the statement that glutamine is "the most important neurotransmitter" should probably be changed, there are of course multiple neurotransmitters that are all absolutely critical.
  • Thank you: “…interactions between one of the most important neurotransmitters…” (page 2, line 74).
  1. Line 88-89: If possible, please cite a reference showing that adrenal 11betaHSD2 expression is capable of dampening cort secretion. Since adrenal Cyp11b1 is orders of magnitude higher than Hsd11b1, which is orders of magnitude higher than Hsd11b2, this does not seem very likely.”
  • Thank for this suggestion. We have included some thoughts within the Discussion section:
  • “The complexity of 11β-HSD action has been highlighted by studies of 11β-HSD1 knockout mice, which display compensatory adrenal hyperplasia and increased basal levels of corticosterone, despite the presumed absence of hepatic cortisone to corticosterone conversion [1]. In addition, 11β-HSD2 provides protection of the target tissues and modulates circulating levels of CORT, as shown in birds [2]. Moreover, peripheral antagonism of the 11β-HSD system has a greater impact on circulating glucocorticoid levels than central control during the stress response, and this system, in turn, could be influenced by ACTH, which caused a 5-10 fold increase in 11beta-HSD2 mRNA in primary cultures of rat adrenocortical cells. [3] The unique advantage of such local regulation is that the adrenal cortex could rapidly and reversibly modify the secreted ratio of inactive to the active form of CORT. A rapid transition of this ratio might serve as an advantage of AFR over APR animals, by preventing negative effects associated with a long-term decrease in GC levels. As the adrenal cortices express both 11βHSD1 and 11βHSD2, we postulate that both activating and inactivating reactions may take place, representing a previously unsuspected rapid regulatory mechanism of the stress response.

  1. Lines 140-145: Were temporal changes in behavior tested statistically? It appears that the results describe changes in the means but have no accompanying analyses to support these conclusions.
  • Thank you for this comment. In over ten days of the experiment, we obtained at each individual point differences between the groups, without setting time as a separate factor. Unfortunately, our capacity to analyse data on the behavioral responses of the animals were limited by the absence of a normal distribution. Therefore, we were forced to apply the Kruskal — Wallis test instead of repeated ANOVA for the dependence of the effects of stress on time.
  1. What rat strain was used? "Genetically similar" is completely vague - please give a more specific description of what this means. Why were only male rats used for this study?
  • Thank you for this comment. We included this information in the Methods section as follows: “Experiments were performed on 29 genetically similar white male Sprague-Dawley rats, each weighing 240–260 g (age of 8-9 weeks)…” (page 9, line 273).

The preference for males was due to two reasons: first, predatory stress is generally viewed as an experimental model of war stress, and, second, because the reproductive cycles of females might introduce variability in the results.

  1. CIOMS has released much more recent "International guiding principles for biomedical research involving animals", at least as recently as 2012, perhaps this would be a more up to date guide to use.”
  • Thank you for this comment. We made the correction in the Methods section.
  • We included the project and number of the study protocol approved by the Committee for Bioethics and Humane Treatment of Laboratory Animals at South Ural State University, Russia.
  •  
  1. Figure 1

- Please label the y-axes on the graphs, this would make the figures much easier to read.

- Error bars below 0 suggest that the data are not normally distributed, perhaps median and quartile ranges would be a more appropriate way to graph the variation in the data? Offsetting the different lines from each other might also help the reader to see the error bars more clearly.

- caption perhaps AOR and PDR should be switched to APR and AFR?”

  • Thank you for this important suggestion. We replaced all pictures with the appropriate changes in each figure.
  1. Figure 3

- this should be labeled Figure 2

- the y-axis reads "Glutamate+Glutamine" but according to the text I think this should be "Glutamate/Glutamine", is that correct? This is also called Glu+Gln in line 174, and should be changed here too.

- perhaps the control group could be the first bar, to be consistent with the ordering of Table 1”

  • Thank you for these suggestions. We replaced all pictures with the changes in each figure. We also refer to “Glu+Gln” or “glutamate metabolites”.
  1. Line 208: Do newborn rats produce cortisone? And are there data showing that 11betaHSD1 is important for this cortisone to promote freezing?
  • Thank you for indicating this error, which we corrected by replacing the phrase “cortisone” by “CORT”.
  1. There is no explanation of why the glutamine/glutamate ratio is used - do this authors interpret this as a greater production of glutamate? Greater metabolic breakdown of glutamine? Some combination of both?
  • Thank you for this insightful comment. We determined only the total content of glutamate and glutamine, as the peaks of these amino acids in the NMR spectrogram were fundamentally inseparable. Thus, NMR spectroscopy cannot identify glutamatergic neurons. Glutamine can be considered both as a metabolite and as a precursor of glutamate. Glutaminase breaks down glutamine to release NH3 and glutamate. Despite the limitations in the identification of glutamate, NMR spectroscopy is valuable in that it allows one to determine the content of glutamine plus glutamate in the striatum of living rats. This provides good opportunities for direct comparison of changes in the striatum of both rats and humans in the study of the neurobiology of stress.

Round 2

Reviewer 3 Report

MAJOR COMMENTS

In my original review I had asked for data validating the steroid extraction and HPLC quantification protocols, since this information is necessary to show the data in Fig 3 are reliable, to support the model given in Fig 4, and to support the conclusions in the abstract and newly added discussion. Validations are still absent. One of the newly added citations (Selyatitskaya et al 2008) gives the extraction procedure but has no validations, and the other citation I am unable to find on Pubmed, Google Scholar, or Sci-Hub. Therefore, I will again ask: How was the method for HPLC steroid quantification validated? Were controls run to demonstrate accurate quantification of tissue steroids? Please provide some examples of HPLC traces (could be in Fig 3 or as supplementary data).

Fig 1 - the results section discusses many changes in behavior over time, which are not statistically analyzed. The authors state in the rebuttal that they were unable to test these because of non-normal data and had to use a Kruskal-Wallis test. There are however many options, such as using a Friedman test, or perhaps normalization of data, that would allow them to explicitly test for changes over time. Without testing these effects, such statements should probably be removed from the results.

MINOR COMMENTS

Introduction line 58-60 - are the authors discussing basal or PSS-induced GC levels? This has not yet been clarified.

Figure 1 - the y-axes must be changed so the full error bars are presented - right now a number of them are cut off.

The authors' description of glutamine+glutamate measurement makes sense. A brief explanation of what they "%" value refers to might be helpful for readers.

The manuscript has two references sections, these should be merged and correct to avoid duplicates.

Line 260 - change "cortisone" to "11-dehydrocorticosterone".

In the fugure, I would recommend also studying females - if these authors are modeling "war stress" as they state, then this is something that certainly also affects females. Additionally, using both greatly strengthens any results.

Round 3

Reviewer 3 Report

The UV traces of adrenal extracts show that adrenal CORT data are available. Since the CORT/DHC ratio is really the most important data to demonstrate a change in adrenal 11betaHSD activity (a conclusion the authors make in this paper), I think this is important to include. Once adrenal CORT data are included I believe this paper is suitable for publication.

Author Response

According to the reviewers comment, we included the CORT/DHC ratios within the results section as follwows: "Moreover, the 11-dehydrocorticosterone/CORT ratio was different (F2,26=3.7; p=0.041) between both groups of stressed rats (AFR and APR). AFR rats (35.04±11.25 n=9) were characterized by lower CORT/11-dehydrocorticosterone ratio than APR (55.16±16.12 n=12; p=0.0006) and control rats (52.33±18.25; n=8; p=0.033), while in APR rats the 11-dehydrocorticosterone/CORT ratio was similar to that of the control (p=0.25)" (line 210-214)